# Diatom Algae-Indicators of Water Quality in the Lower Zarafshan River, Uzbekistan

**Sophia Barinova [1],\* and Karomat Mamanazarova [2]**

[1]  Institute of Evolution, University of Haifa, Abba Khoushi Ave, 199, Mount Carmel, Haifa 3498838, Israel
[2]  Institute of Botany of Uzbek Academy of Sciences, Tashkent 100125, Uzbekistan; karomat.3005@mail.ru
\*  Correspondence: sophia@evo.haifa.ac.il; Tel.: +972-4824-97-99

**Abstract:** This work is the first, the purpose of which was a comprehensive assessment of the ecological state of the lower reaches of the Zarafshan River using bioindication of water quality by diatoms based on species' ecological preferences, pollution indices, statistics, and ecological mapping. A total of 198 species and subspecies of diatoms were first identified from 195 samples collected four times a year at six sites in the lower reaches of the Zarafshan River in 2009–2015. The richest species were *Cymbella*, *Navicula*, and *Nitzschia*. *Pleurosira laevis*, resistant to salinity, was first found in aquatic habitats in Uzbekistan. Bioindicators of nine environmental variables make up 91% of the list. Distribution analysis of variables, pollution indices (SLA), and toxicity indices (WESI) show increases in salinity, turbidity, and decreases in organic pollution downstream. The source of acidification can be the Navoi region. We found an increase in the ability to self-purify with an increase in species richness and abundance of diatoms in the lower part of Zarafshan. Thus, the ecosystem of the studied part of the river successfully copes with the incoming pollution from the middle part of Zarafshan and demonstrates some stability and successful self-purification with a water quality class of 2–3. The first studied lower reaches of the ecosystem of the Zarafshan River using bioindicators, statistics, and ecological mapping show that the problem of aridization in Central Asia does not necessarily lead to degradation of the river ecosystem and an increase in pollution, but with rational water use can improve water quality and self-purification processes. Hence, diatoms can be good indicators of river water quality in a semi-arid region and reflect the climate and anthropogenic load change. We recommend that attention be paid to nutrient and turbidity management and to expand state monitoring points to the lower part of the river up to the Karakul region.

**Keywords:** bioindication; water quality; phytoplankton; phytoperiphyton; Zarafshan River; Uzbekistan

## 1. Introduction

Algae, being mostly autotrophic, form the basis of the trophic pyramid and, therefore, are the first to participate in the production of organic matter in the aquatic ecosystem, using biogenic compounds of nitrogen and phosphorus, as well as dissolved organic matter and play an important role in self-purification of water [1]. The intensity of the biogenic load on water bodies is reflected by the abundance of algae developing under these conditions and their species composition [2,3]. Thus, the algal community's characteristics, such as the abundance, biomass, and species composition, are used in bioindication methods to detect water quality changes [1]. The most developed system of bioindicators is based on diatoms [4]. These methods provide a comprehensive assessment of the results of all processes occurring in a water body. Besides, bioindication using algal communities is a cheap express method, and chemical analysis requires certain financial costs, instrumental base, and relevant legislation [5].

The lower part of Zarafshan River flows through the semi-arid territory of the southern desert of Central Asia [6], where the disappearance of fresh groundwater and unstable expansion of irrigation has largely depleted the rivers [7–9] because of the climate in the region, which change with a tendency towards desertification [7,8,10,11]. The waters of the Zarafshan river basin are used in industry, utilities, and agriculture and, as a result, are polluted from various sources. Thus, all the problems of climate change and anthropogenic pressure affecting water quality in the region are summed up in the lower reaches of the Zarafshan River [10], where only three points of chemical monitoring are presented [12].

When we talk about water quality, it means the physical, chemical, and biological characteristics of water [5]. Water quality determines the functions of ecosystems, social problems, economic growth of communities, and human health [13]. Therefore, water quality assessment should be based on biological and chemical variables recommended by the European Framework Directive (WFD) [14]. Freshwater algae are widely used for ecological assessment of water quality [2,15]. It is important to be aware of algae's diversity in inland waters as most algae species can be used as indicators of the environment. Diatoms usually account for about half of the species richness of well-studied regional mid-latitude algal flora. They are used as good indicators of the environment as the most studied in this regard [2,4].

Analysis of the published data showed that the ecological state's assessment by the diversity of algae in the Zarafshan River for its lower section has not yet been carried out. In previous years, hydrochemical variables and the composition of algae in the Zarafshan River's periphyton are given for the upper and middle parts [16–19]. Several more publications are devoted to studying algae's diversity, including diatoms, in rivers and reservoirs of Uzbekistan [20–28].

The ecological conditions of the rivers of Central Asia's semi-arid regions (high turbidity, anthropogenic load, regulated water consumption, an extensive network of drainage canals for irrigating crops) are unfavorable for developing algae in plankton [18]. However, phytoperiphyton communities are not carried away by the current. Therefore, they are good indicators of water quality changes in rivers [2,15,29–31]; therefore, it is better to take samples of algae inhabiting the water column and underwater substrates. According to current trends [32], water quality assessment should consider the river basin principle, which considers the aquatic ecosystem as an accumulative part of the basin. In this regard, environmental mapping [33] within the framework of the WFD is carried out according to the basin principle, which makes it possible to assess the contribution of the catchment territory to the total pollution of the river following international experience [7,18,32,34]. We hypothesize that bioindication can provide more information on water properties changes along the Zarafshan River than only the chemical data available for the lower Zarafshan at only three sites. Therefore, for environmental assessment, we need to involve chemical data, identify the species composition of diatoms throughout the studied section of the river so that the percentage of indicator species is more than 50%, apply statistical methods of analysis and basinal principle.

This work is the first, the purpose of which was a comprehensive assessment of the ecological state of the lower reaches of the Zarafshan River based on bioindication of water quality by diatoms and ecological mapping.

## 2. Materials and Methods

### 2.1. Description of Study Area

Zarafshan originates at the junction of three mountain systems—Turkmenistan, Zarafshan, and Gissar (Alay) ridges from the Zarafvshan glacier. Individual peaks of the first two ridges exceed 5000 m, and the Gissar one almost reaches 5000 m. The Zarafshan Valley is the largest intermontane depression in Central Asia. Its length is equal to the length of the Zarafshan River—877 km. It is located between the Turkestan ridge in the north and the Zarafshan ridge in the south and occupies the Gissar ridge [32].

The main part of the Zarafshan River's water is irrigating the Samarkand, Navoi, and Bukhara regions. River water is partially absorbed by the sands of the southwestern spurs of the Kyzyl Kum. In the 1960s–1970s. An insignificant part of the river water previously reached Lake Dengizkul [11].

For the efficient use of the waters of Zarafshan, the Kattakurgan and Kuyimazar reservoirs, as well as several hydroelectric facilities, have been built.

Beyond Penjikent, Zarafshan crosses the border with Uzbekistan. At the crossing of the border between Tajikistan and Uzbekistan on Zarafshan, there is the Rawat-Khoja dam. The big Dargom canal begins here. Below the dam, Zarafshan forms an oasis, in which Samarkand is located along the river.

By its geomorphological nature, the Zarafshan river basin is divided into 3 parts: upper, middle, and lower reaches. The upper part course belongs to the mountainous region on the territory of Tajikistan, the middle (193 km)—to the foothill zone of the Samarkand region of Uzbekistan, the lower (287 km)—to the flat zone of the Navoi and Bukhara regions [8,10] (Figure 1). Lower studied part of the Zarafshan River basin is different in climatic properties. The climate in the region of Bukhara is semi-arid, with an average annual temperature of 15.0 °C and precipitation of about 156 mm per year. There is absent summer rainfall, but in March, it an average of 33 mm. The average temperatures vary between 28.8 °C in July and 0.8 °C in January. A significant property of climate in the region is that precipitation is below potential evapotranspiration [35]. Average air temperature and average annual precipitation have opposite tendencies down the river and are especially different in the lower part of Zerafshan near Bukhara [19].

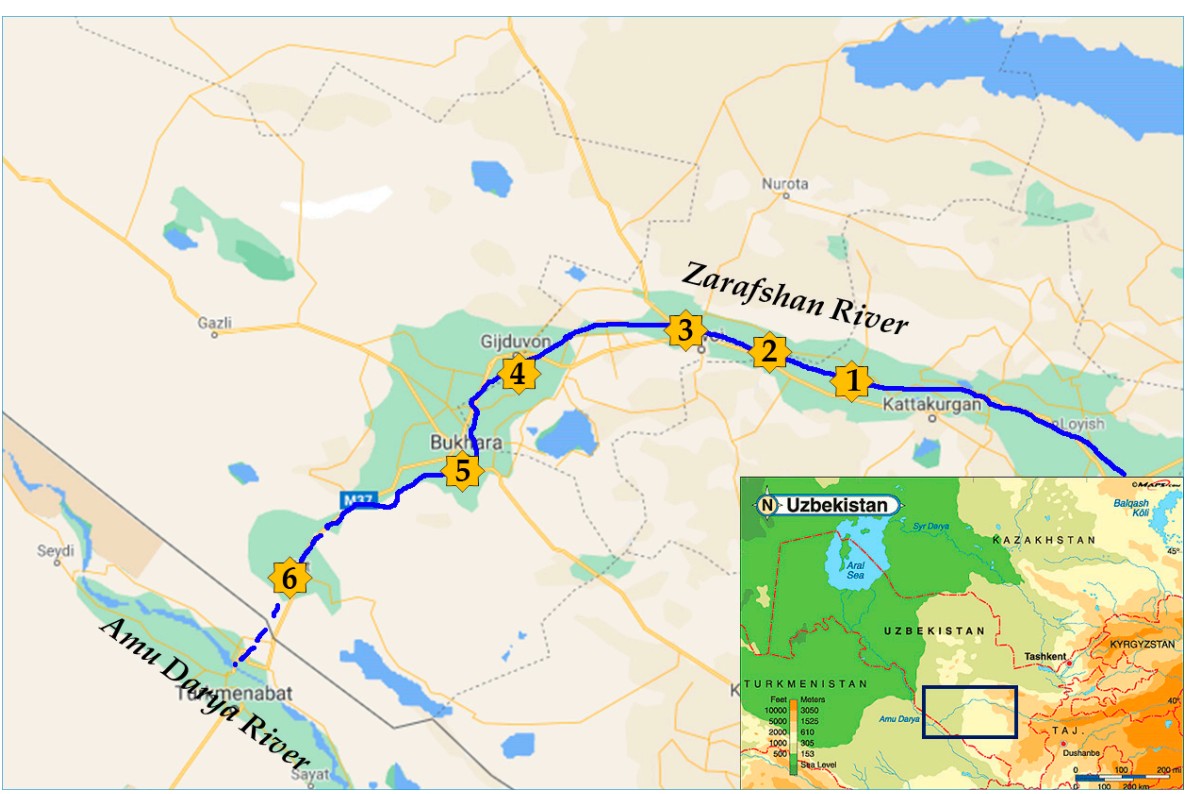

**Figure 1.** Map of diatom sampling points (orange stars numbered as in Table 1) in the Zarafshan River, 2009–2015.

**Table 1.** Sampling points with GIS coordinates, distance from upper site, environmental and averaged biological variables in the Zarafshan River, 2009–2015. No.—site number. SLA—Sládeček index of saprobity.

| Site | No. | Code | North | East | Altitude, m a.s.l. | Distance, km | Number of Species | Sum of Scores | SLA |
|------|-----|------|-------|------|-------------------|--------------|-------------------|---------------|-----|
| Khatirchi | 1 | Khat | 40.00.32 | 65.57.10 | 415 | 0 | 38 | 114 | 1.55 |
| Pakhtakor | 2 | Pakh | 40.05.53 | 65.39.26 | 376 | 32.11 | 30 | 82 | 1.47 |
| Navoi | 3 | Navo | 40.09.35 | 65.19.43 | 329 | 34.35 | 52 | 154 | 1.66 |
| Gizhduvon | 4 | Gizh | 40.03.40 | 64.46.25 | 264 | 54.74 | 32 | 88 | 1.45 |
| Bukhara | 5 | Bukh | 39.49.41 | 64.23.17 | 224 | 48.47 | 73 | 197 | 1.53 |
| Karakul | 6 | Kara | 39.30.39 | 63.49.03 | 196 | 63.12 | 84 | 249 | 1.41 |

*2.2. Field Sampling*

The material for the work comes from algological samples collected in 2009–2015. Samples were collected at 6 permanent stations located in the lower reaches of the Zarafshan River (Figures 1 and 2) in four seasons. The collection of samples and their processing were carried out according to generally accepted algology methods [36–39]. Plankton samples were taken with a silk gas plankton net (No. 78). Periphyton samples were collected with a scalpel or scraping with a knife, removed from aquatic vegetation and dead plant substrate at a depth of 0 to 50 cm along the banks of the river, from the surface of underwater objects in shallow places and the contact zone of water-silt from a certain area (10 cm²). Benthos samples were taken from the bottom of the river using improvised means. Fixation with 4% neutral formaldehyde solution was carried out immediately after collection according to the standard technique [37]. During the study period, 195 algological samples were collected and processed. During the collection of materials, the air and water temperature, the river's width, the speed of the water flow, color, and transparency, the pH value, and visualized sources of water pollution were determined. We collected our samples in the same monitoring points in parallel with UzHYDROMET [12] (2009–2012), so chemical data was taken from this recourse in which the chemical variables value was determined according to [40]. Coordinate and altitude referencing of the stations was done by Garmin eTrex GPS-navigator (Table 1).

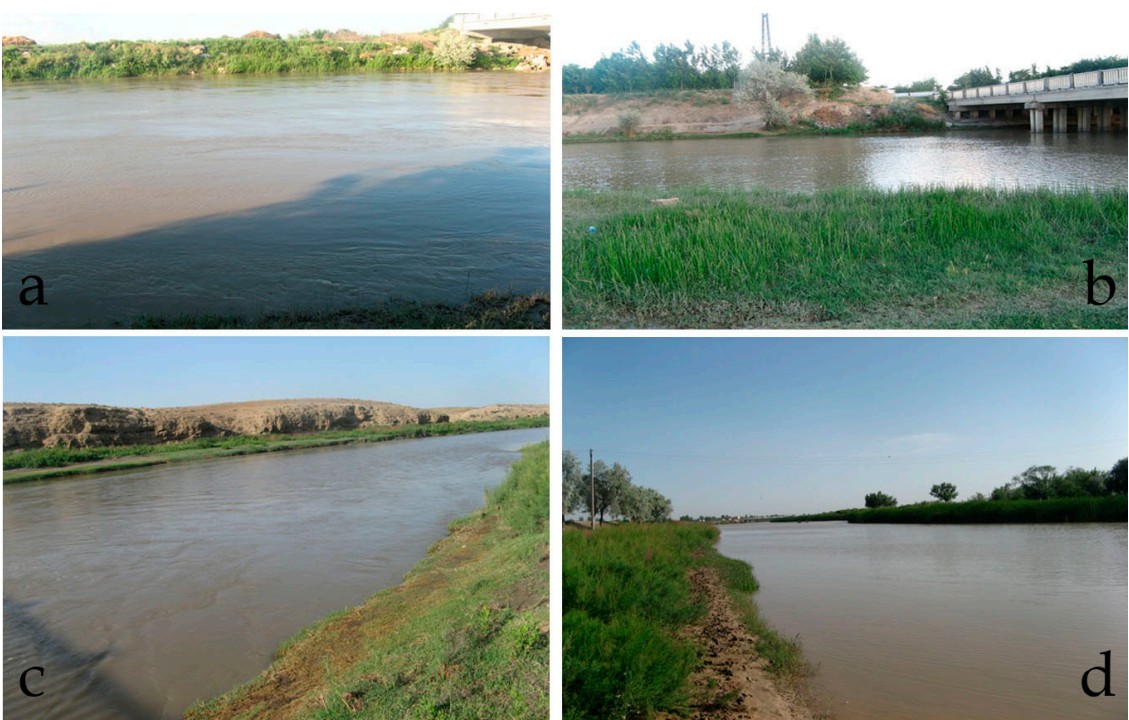

**Figure 2.** Sampling sites in the Zarafshan River: Khatyrchi (**a**), Navoi (**b**), Ghidzhuvan (**c**), Bukhara (**d**).

### 2.3. Laboratory Processing

The fixed material was processed in the laboratory using a Carl Zeiss microscope, at a magnification of ×150–×1000 (MI) with a 40 and 100× oil-immersion plan-apochromat objective with a 1.25 aperture. In the process of identifying some diatoms, electron microscopic methods were used. The studies were carried out on a Quanta 250 scanning electron microscope at the Department of Electron Microscopy of the Zoological Institute of the Russian Academy of Sciences (St.-Petersburg) in the Laboratory of parasitology. The data was saved as digital images. Morphometric data were obtained by measuring the shells. Preparation of the permanent slides and preparations for a scanning microscope was carried out by the peroxide method [41]. For purification of the material from the organic content, the diatom sample drop was put on a cover glass slide; a solution of 40% hydrogen peroxide ($H_2O_2$) was added a few times on the drop of sample and dried with heating. Washed with distillate several times [41]. The relevant handbooks, monographs, and individual articles were used [42–58] to determine species. The list of revealed diatoms in the Zarafshan River was updated with an algaebase.org [59].

The total abundance was found for each species of planktonic and periphyton diatom species as abundance scores [41]. The total score abundance of algae in the community was determined by summing up all species abundance scores found in the sample. Bioindication properties of each revealed species come from our world database [30,31]. Indicators of the water temperature range were divided into four categories on the base of collected information and the temperature ranges of species occurrence [3] where the category "warm" included species that survived in the water of 20–40 °C with an optimum between 27.5–30.0 °C; "eterm" in 0–40 °C, with optimum about 20.0 °C; "temp" in 0–40 °C, with an optimum between 14–26 °C; and "cool" survived in the water below 14 °C.

The saprobic index (SLA) was calculated according to V. Sládeček [60], according to Pantle and Buck's [61] method to estimate the level of organic pollution. Index values SLA ranges from 0 (no polluted) to 4.5 (very polluted) for the aquatic environment. All data were ranked according to the CIS countries' classification system [1] to assess the lower part of the Zarafshan River basin water quality.

The Water Ecosystem State Index (WESI) [1] was calculated using the equation: WESI = Rank Index SLA/Rank N-$NO_3$ to assess the toxic pollution of the Zarafshan River. The index values vary from 0 to 9. If the index values are less than one, then the ecosystem is exposed to toxic pollution inhibiting photosynthesis.

The River Pollution Index (RPI), according to Sumita [30,62], was calculated for chemical and biological data for sampling sites and the distance between them. RPI of Sumita was created for the Watanabe index of pollution only. Earlier, we proposed the calculation of integral indices to assess the organic pollution of water in the river and the chemical variables of the water also [30]. The river pollution index (RPI) is a constant value with constant loads on a water body. It can be proposed as a passport value for a water body [62]. The relative stability of the integral indices noted by Sumita is also realized in this case. RPI is calculated as the integral of the indicator values relative to the sections along the river's length using the formula:

$$RPIv = \Sigma\,(Vi + Vj) \times l/2L,$$

where:

Vi, Vj—variable value for adjacent stations i, j;
l—distance between two adjacent stations (km);
L—total length of the river between the first and last station.

The calculated index value can be classified on the same scale as the particular values of the variable.

Canonical Correspondence Analysis was done for revealing relationships between biological and environmental variables with the CANOCO 4.5 Program [63]. Calculation

of similarity was doing as the network analysis in JASP (significant only) on the botnet package in R Statistica [64].

## 3. Results

### 3.1. Physico-Chemical Characteristics of the Lower Part of the Zarafshan River

Table 1 shows that the river flows in the flat landscape where the altitude decrease from 415 to 196 m above sea level (a.s.l.). The distance between sampling sites is an average of about forty km. The river width varied between 50–100 m. Its banks are ruderal or with low developed vegetation. According to the chemical data, the water is fresh, slightly alkaline, and soft (Table 2). We have chemical information from three upper monitoring points only. In this part of the river can be seen that oxygenation of the river water decreases together with Zn, As, phenols and detergents. However, all other environmental variables are increased down the river flow. Especial attention can be given to the TSS of water, the high value of which suppresses the photosynthetic activity. In the same direction, increased COD means that oxygen spent on the chemical matters oxidization. Can be seen the increasing copper, fluorine, chrome, and iron together with nitric forms of nutrients, BOD, and alpha-GHCG. Whereas DDT and gamma-GHCG were not revealed.

**Table 2.** Range and average value of hydrochemical variables and index RPI in 3 sites of the Zarafshan River, 2009–2015.

| Variable | Khatirchi | Average | Pakhtakor | Average | Navoi | Average | Index RPI |
|---|---|---|---|---|---|---|---|
| $O_2$ mg $L^{-1}$ | 5.2–9.54 | 7.295 | 6.8–8.79 | 7.820 | 5.07–8.44 | 6.983 | 7.477 |
| BOD, mgO $L^{-1}$ | 0.77–2.7 | 1.638 | 1.36–3.62 | 2.155 | 1.83–3.8 | 2.860 | 2.212 |
| COD, mgO $L^{-1}$ | 6.75–19.7 | 12.438 | 16.67–35.2 | 26.100 | 19.12–40.9 | 31.125 | 24.098 |
| $N-NH_4$, mg $L^{-1}$ | 0.02–0.07 | 0.043 | 0.06–0.43 | 0.210 | 0.10–0.25 | 0.165 | 0.158 |
| $N-NO_2$, mg $L^{-1}$ | 0.01–0.035 | 0.025 | 0.019–0.129 | 0.053 | 0.063–0.178 | 0.120 | 0.064 |
| $N-NO_3$, mg $L^{-1}$ | 1.09–5.57 | 2.688 | 1.84–3.47 | 2.463 | 2.33–8.79 | 4.830 | 3.129 |
| Fe, mg $L^{-1}$ | 0.01–0.07 | 0.030 | 0.03–0.14 | 0.073 | 0.04–0.18 | 0.090 | 0.067 |
| Cu, mcg $L^{-1}$ | 2.6–8.9 | 5.250 | 3.7–8.0 | 5.400 | 4.0–8.2 | 6.100 | 5.545 |
| Zn, mcg $L^{-1}$ | 2.5–9.7 | 5.650 | 3.1–5.3 | 3.700 | 3.2–5.5 | 4.275 | 4.320 |
| Phenols, mg $L^{-1}$ | 0.001–0.009 | 0.005 | 0.001–0.003 | 0.001 | 0.001–0.005 | 0.002 | 0.002 |
| Oil, mg $L^{-1}$ | 0–0.05 | 0.020 | 0–0.01 | 0.005 | 0–0.09 | 0.030 | 0.015 |
| Detergents, mg $L^{-1}$ | 0–0.05 | 0.015 | 0–0.07 | 0.020 | 0–0.02 | 0.005 | 0.015 |
| TSS, mg $L^{-1}$ | 198.8–1118 | 591 | 396.9–1250 | 1434 | 200.6–2316 | 1028.1 | 1125.458 |
| DDT, mg $L^{-1}$ | 0 | 0 | 0 | 0 | 0 | 0 | 0.000 |
| Alpha-HCCG, mcg $L^{-1}$ | 0 | 0 | 0.001–0.012 | 0.005 | 0.003–0.004 | 0.002 | 0.003 |
| Gamma-HCCG, mcg $L^{-1}$ | 0 | 0 | 0 | 0 | 0 | 0 | 0.000 |
| Cr VI, mcg $L^{-1}$ | 0.3–1.5 | 0.825 | 0.6–1.8 | 1.200 | 0.8–1.8 | 1.275 | 1.129 |
| F, mg $L^{-1}$ | 0.52–1.0 | 0.693 | 0.53–0.96 | 0.780 | 0.63–1.02 | 0.865 | 0.781 |
| As, mcg $L^{-1}$ | 0–11.0 | 3.125 | 0 | 0 | 0 | 0 | 0.755 |
| TDS, mg $L^{-1}$ | 537.7–878.1 | 728.8 | 1245.4–1989.5 | 1591.9 | 1330.7–2014.5 | 1710.8 | 1414.124 |

### 3.2. Biological Characteristics of the Lower Part of the Zarafshan River

Altogether 198 taxa belonged to 62 Genera of diatoms were revealed from 195 samples collected during 2009–2015 in the Zarafshan River from six sampling sites (Appendix A). The most represented in the list are genera *Cymbella* with 16 taxa and *Navicula* and *Nitzschia* with 15 taxa of each. *Gomphonema* represents 9 taxa, but genera *Amphora*, *Sellaphora*, *Surirella*, and *Cyclotella* by 7 taxa of each. All other genera in the list included 1 to 6 taxa, and only 30 genera were monospecific. The most species-rich community was in the lower site Karakul (Table 1), with 84 taxa. As can be seen in Table 1, species richness is increased from the upper studied site Khatirchi to lower site Karakul up to three times.

Cell abundance has the same trend in distribution, increasing from upper to lower site (Table 1). The most abundant (dominant) species in site Khatirchi were *Cavinula lacustris*, *Pleurosira laevis*, *Brachysira microcephala*, and *Navicula rostellata*. Site Pakhtakor communities were represented by the well abundant species *Pleurosira laevis*. In communities of the Navoi site *Rhoicosphenia abbreviata*, *Synedra famelica*, and *Navicula rostellata* were dominant species. Communities in the site Gizhduvon have not some abundant taxa, whereas followed site Bukhara abundant species were *Crenotia thermalis*, *Nitzschia angularis*, *Nupela neogracillima*, *Cymbella cymbiformis*, *Navicula rostellata*, and *Sellaphora wummensis*. In the lower site Karakul, diatom communities were represented by abundant species *Achnanthes dispar* var. *angustissima*, *Cocconeis placentula* var. *euglypta*, *Diploneis smithii* var. *pumila*, *Gomphonema tergestinum*, *Sellaphora mutata*, *Caloneis bacillum*, *Navicula rostellata*, and *Pantocsekiella rossii*. Therefore, only two species (*Pleurosira laevis* in two upper sites and *Navicula rostellata* in four sites across the river) were abundant in communities more than one site, but all other communities were formed under the site-specific condition.

A floristic comparison of the similarity of diatom communities from six sites in the Zarafshan River is presented in Figure 3. The JASP network plot shows that the upstream Khatirchi and Pakhtakor communities were most similar (significant only, *p* < 0.05). The communities of the Navoi and Gizhduvon sites form the next core of similarity. The correlation between all other sites' communities is negative. This means a high specificity of communities in the areas from Navoi to Karakul.

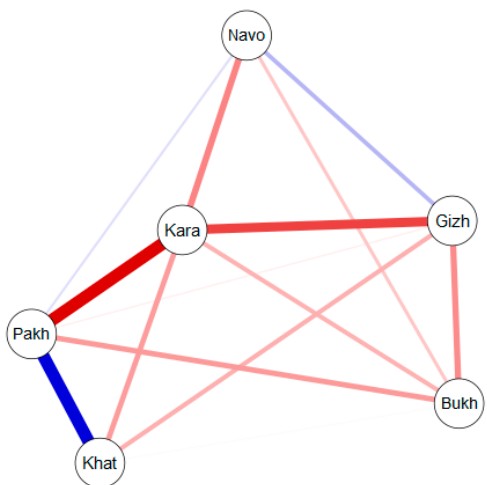

**Figure 3.** The JASP network plot of the similarity (in R-statistics) of diatom communities from six sites in the Zarafshan River, 2009–2015. The names of sampling sites as in Table 1. The line thickness between sites reflect the correlation value, blue is positive, red is negative.

### 3.3. Bioindicators in the Lower Part of the Zarafshan River

Bioindication results are represented in Figures 4–6 and Table 3, with the percent of each ecological group's indicator taxa on six studied site communities. Diatom species were divided into three groups in their substrate preference in the aquatic environment (Figure 4a). The percent of benthic species increased down the river with decreasing at the same time in planktonic inhabitants. Water mass in the river is warmer down the stream with increasing eurythermic and warm-water indicators (Figure 4b, Appendix A, Table 3). Water oxygen indicators show increasing in low-oxygenated water indicators down the river course (Figure 4c). At the same time, the influence of acidification can be seen from site Navoi and up to lower site Karakul by increasing the percent of acidophilic indicators (Figure 4d).

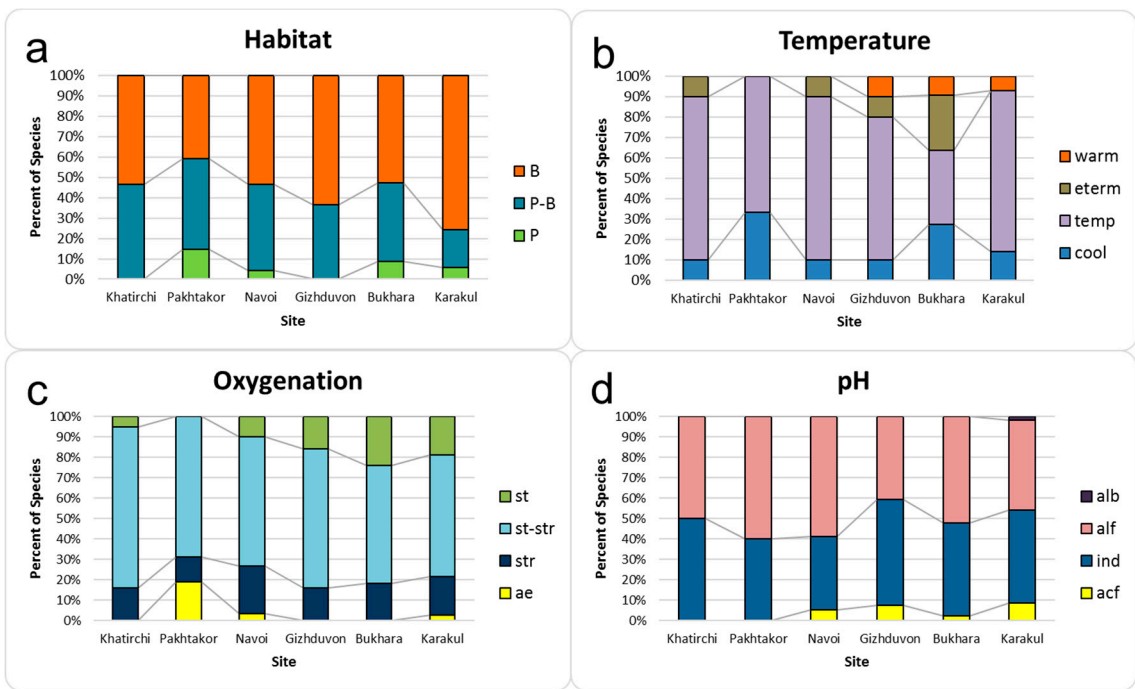

**Figure 4.** Distribution of bioindicators of habitat preferences, temperature, oxygenation, and pH in the Zarafshan River, 2009–2015. Abbreviation for ecological indicator groups: (**a**), Habitat preferences (Hab): B, benthic; P-B, planktonic-benthic; P, planktonic. (**b**), Water temperature (T): cool, cool-loving species; temp, temperate temperature water inhabitants; eterm, eurythermic species, warm, warm water inhabitants. (**c**), Streaming and Oxygenation (Oxy): aer, aerophiles, str, streaming waters inhabitant; st-str, low streaming waters inhabitant; st, standing water inhabitant. (**d**), Water pH (pH): acf, acidophilic species; ind, indifferent; alf, alkaliphilic species; alb, alkalibiontes.

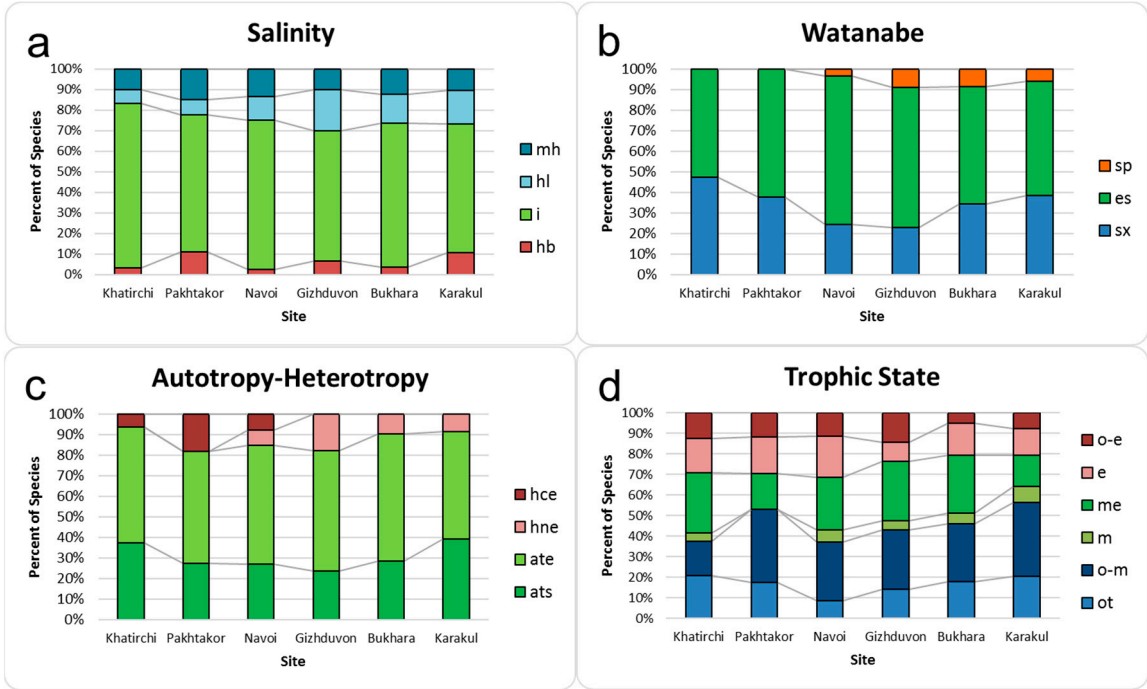

**Figure 5.** Distribution of bioindicators of salinity preferences, organic pollution according to Watanabe, nutrition type, and trophic state in the Zarafshan River, 2009–2015. Abbreviation for indicator groups: (**a**), Water salinity (Sal): hb, halophobe; i, oligohalobious-indifferent; hl, oligohalobious-halophilous; mh, mesohalobious. (**b**), Organic pollution, Watanabe (D): sx, saproxenes, es, eurysaprobes; sp, saprophiles. (**c**), Nutrition type as Nitrogen uptake metabolism (Aut-Het): ats, nitrogen-autotrophic taxa, tolerating very small concentrations of organically bound nitrogen; ate, nitrogen-autotrophic taxa, tolerating elevated concentrations of organically bound nitrogen; hne, facultatively nitrogen-heterotrophic taxa,

needing periodically elevated concentrations of organically bound nitrogen; hce, nitrogen-heterotrophic taxa, needing elevated concentrations of organically bound nitrogen. (**d**), Trophic state (Tro): ot, oligotrafentic; o-m, oligo-mesotraphentic; m, mesotraphentic; me, meso-eutraphentic; e, eutraphentic; o-e, oligo- to eutraphentic.

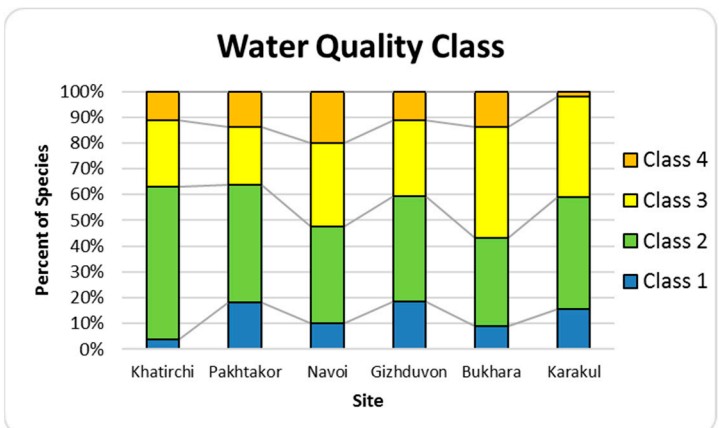

**Figure 6.** Distribution of bioindicators of water quality on the base of species-specific indices of organic pollution according Sládečec in the Zarafshan River, 2009–2015. Legend: Class of Water Quality 1–5.

**Table 3.** Percent of bioindicator groups in each studied site community of the Zarafshan River, 2009–2015. The ecological groups' order in respect of each group indicator value increasing.

| Indicator Group | Khatirchi | Pakhtakor | Navoi | Gizhduvon | Bukhara | Karakul |
|---|---|---|---|---|---|---|
| **Habitat** | | | | | | |
| P | 0.0 | 14.8 | 4.4 | 0.0 | 8.8 | 5.7 |
| P-B | 46.7 | 44.4 | 42.2 | 36.7 | 38.6 | 18.6 |
| B | 53.3 | 40.7 | 53.3 | 63.3 | 52.6 | 75.7 |
| **Temperature** | | | | | | |
| cool | 10.0 | 33.3 | 10.0 | 10.0 | 27.3 | 14.3 |
| temp | 80.0 | 66.7 | 80.0 | 70.0 | 36.4 | 78.6 |
| eterm | 10.0 | 0.0 | 10.0 | 10.0 | 27.3 | 0.0 |
| warm | 0.0 | 0.0 | 0.0 | 10.0 | 9.1 | 7.1 |
| **Oxygen** | | | | | | |
| st | 5.3 | 0.0 | 10.0 | 15.8 | 24.2 | 18.9 |
| st-str | 78.9 | 68.8 | 63.3 | 68.4 | 57.6 | 59.5 |
| str | 15.8 | 12.5 | 23.3 | 15.8 | 18.2 | 18.9 |
| ae | 0.0 | 18.8 | 3.3 | 0.0 | 0.0 | 2.7 |
| **pH** | | | | | | |
| acf | 0.0 | 0.0 | 5.1 | 7.4 | 2.1 | 8.5 |
| ind | 50.0 | 40.0 | 35.9 | 51.9 | 45.8 | 45.8 |
| alf | 50.0 | 60.0 | 59.0 | 40.7 | 52.1 | 44.1 |
| alb | 0.0 | 0.0 | 0.0 | 0.0 | 0.0 | 1.7 |
| **Salinity** | | | | | | |
| hb | 3.3 | 11.1 | 2.3 | 6.7 | 3.5 | 10.4 |
| i | 80.0 | 66.7 | 72.7 | 63.3 | 70.2 | 62.7 |
| hl | 6.7 | 7.4 | 11.4 | 20.0 | 14.0 | 16.4 |
| mh | 10.0 | 14.8 | 13.6 | 10.0 | 12.3 | 10.4 |
| **Saprobity Watanabe** | | | | | | |
| sx | 47.4 | 37.5 | 24.1 | 22.7 | 34.3 | 38.2 |
| es | 52.6 | 62.5 | 72.4 | 68.2 | 57.1 | 55.9 |
| sp | 0.0 | 0.0 | 3.4 | 9.1 | 8.6 | 5.9 |

| Water Quality Class with EU color code and species-specific SLA ranges | | | | | | |
|---|---|---|---|---|---|---|
| Class 1 (0.0–05) | 3.7 | 18.2 | 10.0 | 18.5 | 9.1 | 15.7 |
| Class 2 (0.5–1.5) | 59.3 | 45.5 | 37.5 | 40.7 | 34.1 | 43.1 |
| Class 3 (1.5–2.5) | 25.9 | 22.7 | 32.5 | 29.6 | 43.2 | 39.2 |
| Class 4 (2.5–3.5) | 11.1 | 13.6 | 20.0 | 11.1 | 13.6 | 2.0 |
| **Trophy** | | | | | | |
| ot | 20.83 | 17.65 | 8.57 | 14.29 | 17.95 | 20.51 |
| o-m | 16.67 | 35.29 | 28.57 | 28.57 | 28.21 | 35.90 |
| m | 4.17 | 0.00 | 5.71 | 4.76 | 5.13 | 7.69 |
| me | 29.17 | 17.65 | 25.71 | 28.57 | 28.21 | 15.38 |
| e | 16.67 | 17.65 | 20.00 | 9.52 | 15.38 | 12.82 |
| o-e | 12.50 | 11.76 | 11.43 | 14.29 | 5.13 | 7.69 |
| **Trophy main groups** | | | | | | |
| ot | 37.5 | 52.9 | 37.1 | 42.9 | 46.2 | 56.4 |
| me | 33.3 | 17.6 | 31.4 | 33.3 | 33.3 | 23.1 |
| e | 29.2 | 29.4 | 31.4 | 23.8 | 20.5 | 20.5 |
| **Nutrition type** | | | | | | |
| ats | 37.5 | 27.3 | 26.9 | 23.5 | 28.6 | 39.1 |
| ate | 56.3 | 54.5 | 57.7 | 58.8 | 61.9 | 52.2 |
| hne | 0.0 | 0.0 | 7.7 | 17.6 | 9.5 | 8.7 |
| hce | 6.3 | 18.2 | 7.7 | 0.0 | 0.0 | 0.0 |

The distribution of species-indicators of chloride concentration reflects increasing water salinity (Figure 5a) with increasing halophilic taxa and mesohalobes. Increasing organic pollution can be seen from site Navoi up to Karakul with eurysaprobic indicators in the diatom communities (Figure 5b). Algae in the lower part of the Zarafshan River were represented by all four ecological groups of the type of nutrition (Figure 5c). Remarkable that facultative heterotrophic species (hce, nitrogen-heterotrophic taxa, needing elevated concentrations of organically bound nitrogen) were presented in the upper three sites decreased down the river with increasing autotrophic taxa up to 90% in Karakul. Six ecological groups that covered the spectrum of trophic indicators from oligo- to eutrophic presented the trophic state's indicators. Figure 5d shows the increase of oligotrophic taxa from site Navoi down the river course with decreasing of eutrophic species at the same time. The main groups of trophic indicators as oligotrophic, mesotrophic, and eutrophic color here with the same colors with different shades. It helps to understand better the decreasing of the trophic level of the Zarafshan River in its lower part.

Of particular interest is usually the water quality class. Figure 6 shows that the identified species-indicators of organic pollution correspond to four water quality classes, according to species-specific SLA indices (Appendix A). Here can be seen that the percentage of indicators of polluted waters of class 4 decreased down the river. In contrast, the percentage of indicators of moderately polluted and unpolluted waters of classes 1–3 increased simultaneously.

At the same time, abundant species show the preferences of organic pollution (as species-specific index SLA) increasing down the river with a succession of indicators from sLA about 1.2 on the upper sites to sLA about 1.9 on the lower (Table 4).

**Table 4.** Distribution of abundant species indicators of organic pollution according the increasing of species-specific index sLA in the Zarafshan River, 2009–2015. Abundance scores and species-specific sLA value are toned in respect of the Water Quality Class ranges (Table 3).

| Taxa | Khatirchi | Pakhtakor | Navoi | Gizhduvon | Bukhara | Karakul | sLA |
|---|---|---|---|---|---|---|---|
| *Cymbella cymbiformis* C.Agardh | 0 | 0 | 0 | 0 | 5 | 0 | 2.00 |
| *Rhoicosphenia abbreviata* (C.Agardh) Lange-Bertalot | 0 | 0 | 5 | 3 | 0 | 3 | 1.90 |
| *Sellaphora mutata* (Krasske) Lange-Bertalot | 0 | 0 | 0 | 0 | 3 | 5 | 1.90 |
| *Sellaphora wummensis* J.R.Johansen. | 0 | 0 | 0 | 0 | 5 | 0 | 1.90 |
| *Caloneis bacillum* (Grunow) Cleve | 0 | 0 | 0 | 0 | 0 | 5 | 1.30 |
| *Cocconeis placentula* var. *euglypta* (Ehrenberg) Grunow | 1 | 0 | 0 | 0 | 0 | 5 | 1.30 |
| *Gomphonema tergestinum* (Grunow) Fricke | 0 | 0 | 0 | 0 | 3 | 5 | 1.30 |
| *Synedra famelica* Kützing | 0 | 0 | 5 | 0 | 0 | 0 | 1.30 |
| *Nupela neogracillima* Kulikovskiy & Lange-Bertalot | 0 | 0 | 0 | 0 | 5 | 0 | 1.20 |
| *Brachysira microcephala* (Grunow) Compère | 5 | 0 | 0 | 0 | 0 | 0 | 1.00 |
| *Lacustriella lacustris* (W.Gregory) Lange-Bertalot & Kulikovskiy | 5 | 0 | 0 | 0 | 0 | 0 | 1.00 |
| *Pleurosira laevis* (Ehrenberg) Compère | 5 | 5 | 0 | 0 | 0 | 0 | 1.00 |
| *Crenotia thermalis* (Rabenhorst) Wojtal | 0 | 0 | 0 | 0 | 5 | 0 | 0.30 |

The comparison of Figure 6 (or Table 3 in percentage) and Table 4 demonstrates that the full picture of the organic pollution dynamic was different from assessing it based on abundant species. It shows the great role of species richness in the diatom community in the self-purification process of the Zarafshan River.

*3.4. Indices of Saprobity (SLA) and Ecosystem State (WESI), and River Pollution Index (RPI) in the Lower Part of the Zarafshan River*

Indication of organic pollution was doing by the indices saprobity SLA (Table 1). We calculated it based on species-specific index sLA value and the abundance scores in each site community (Appendix A). Table 1 shows that SLA indicators correspond to water quality classes 2–3 in the entire studied part of the river. The water quality map has been constructed based on this assessment, as recommended in EU FWD [14,34]. Figure 7 show that water quality fluctuated down the river. It can help divide the studied river course into three different parts with their own organic pollution level. Seen that the upper site Khatirchi receives water with organic matter from the middle part of the river. Then, up to Pakhtakor site, can be seen increasing water quality up Class 2. Nevertheless, the river water received the next part of organic pollution from site Pahtakor to Navoi, and quality decreased to Class 3. From Navoi to Gizhduvon, the water was better than from Gizhduvon to Bukhara, where there was a decrease in Class 3. The last part of the studied river course correspond to Class 2 of water quality and demonstrate self-purification of the river water because water quality arise with changing from Class 3, middle polluted to Class 2, low polluted when assessed by the index of organic pollution SLA (Table 3).

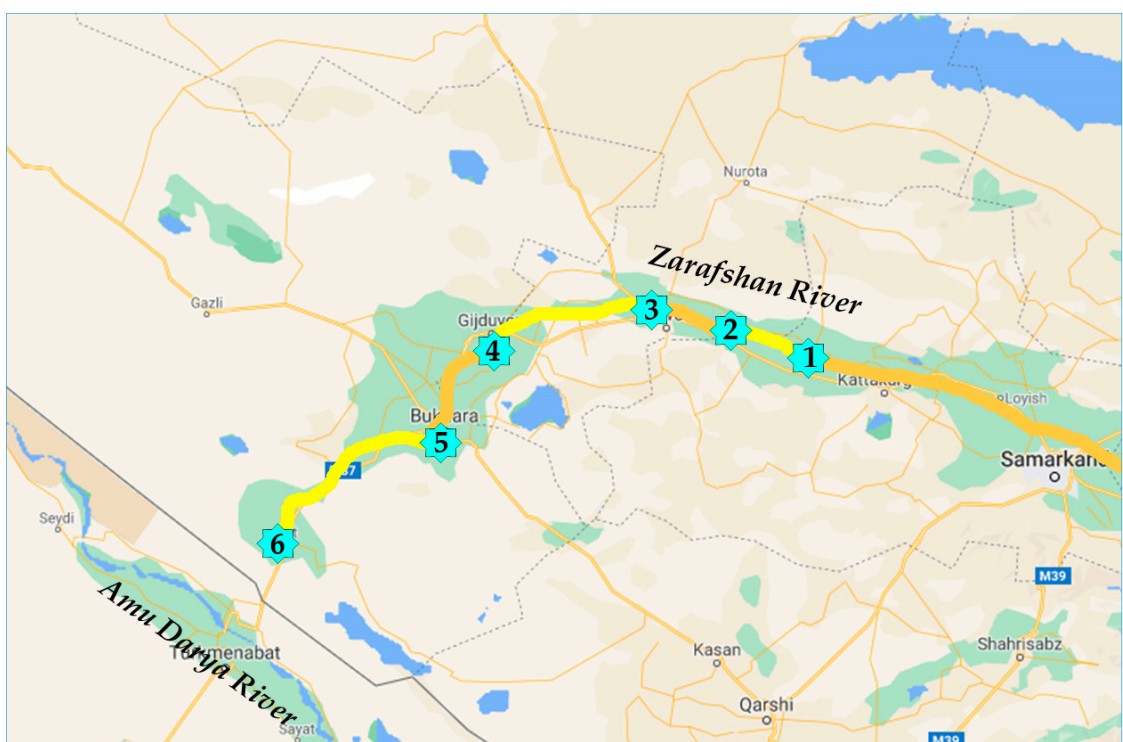

**Figure 7.** Water Quality map in the Zarafshan River, 2009–2015 on the base of Indices saprobity according Sládeček. Color lines as in EU color codes of the Water Quality Classes: yellow—Class 2; orange—Class 3.

The Aquatic Ecosystem State Index (WESI) was calculated based on the saprobity index (SLA) (Table 1) and nitric-nitrogen concentration (Table 2). Figure 8 shows WESI values below unity at the top three sampling points. This made it possible to assume the toxic influence of the river environment on the photosynthesis of diatom communities in the upper reaches of the Khatyrchi-Pakhtakor-Navoi with a tendency to a decrease in the index, which means a slight increase in the toxic influence.

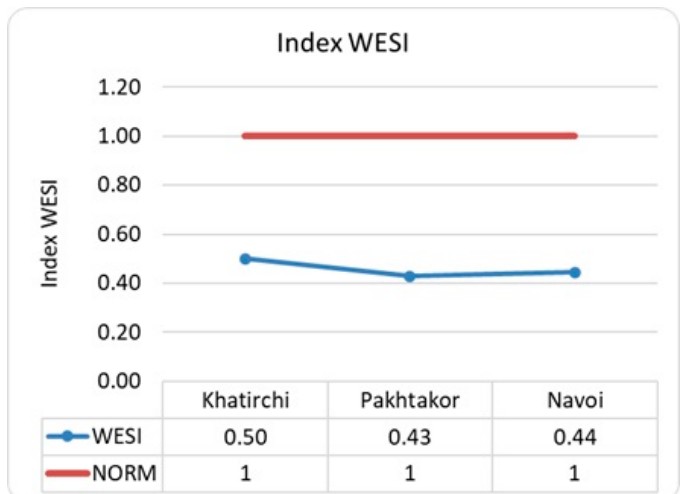

**Figure 8.** Water Ecosystem State Index (WESI) in the Zarafshan River, 2009–2015 on the base of Indices saprobity according Sládeček (SLA) and Nitric-nitrogen concentration. Norm line is the critical value of WESI indicated start of toxic impact if WESI value is below 1.

The River Pollution Index (RPI), according to Sumita, was calculated for Index Saprobity (SLA) (Table 1), WESI, and the biological variables (Table 2). Table 5 shows RPI's

values for biological variables on the sampling sites 1–3 on which were defined the chemical variables value (the first row) and for the whole set of sites 1–6 (second row). Seen that species number in communities was lower in three upper sites than in the entire studied river. Abundance scores for three upper sites and six river sites tend to increase down the river. In contrast, the indices' saprobity SLA dynamic demonstrated the opposite trend and decreasing downstream of the river.

**Table 5.** River Pollution Index (RPI) for biological variables from 1–3 sampling sites and for 1–6 sites in the Zarafshan River, 2009–2015.

| Index RPI | Index WESI | No of Species | Sum of Scores | Index SLA |
|---|---|---|---|---|
| RPI-1-3 sites | 0.45 | 37.62 | 108.34 | 1.54 |
| RPI-1-6 sites | - | 52.83 | 149.52 | 1.51 |

*3.5. Species-Environment Relationships the Lower Part of the Zarafshan River*

Canonical Correspondence Analysis (CCA) was conducted to identify the relationship between species and the environment. We have a set of environment variables for only the upper three sites (Table 2); we took their average values for analysis. The biological part of the calculation also included variables from the same three sites (Table 1, Appendix A). Since mass species determine the face of a community and develop in an optimal environment for themselves, they can adequately show which environmental parameters affect a community's development. Therefore, for the CCA analysis, we selected from the three upper sites' communities only abundant species with scores above 4 from Appendix A, which were identified in 95 samples. Table 6 included coded taxa names, abundance scores on three upper sites, and indicator properties of abundant taxa.

**Table 6.** Abundant taxa from three upper sites (1-2-3) with coded names, abundance scores and indicator properties.

| Taxa | Code | 1 | 2 | 3 | Hab | T | Oxy | pH | Sal | D | Sap | SLA | Tro | Aut-Het |
|---|---|---|---|---|---|---|---|---|---|---|---|---|---|---|
| *Brachysira microcephala* (Grunow) Compère | BraMic | 5 | 0 | 0 | B | - | - | - | - | - | o | 1.0 | o-m | - |
| *Lacustriella lacustris* (W.Gregory) Lange-Bertalot & Kulikovskiy | LacLac | 5 | 0 | 0 | B | - | - | ind | i | - | o | 1.0 | o-m | ats |
| *Navicula rostellata* Kützing | NavRos | 5 | 0 | 5 | B | - | - | - | - | - | - | - | e | - |
| *Pleurosira laevis* (Ehrenberg) Compère | PleLae | 5 | 5 | 0 | B | temp | - | alf | mh | - | o | 1.0 | e | - |
| *Rhoicosphenia abbreviata* (C.Agardh) Lange-Bertalot | RhoAbb | 0 | 0 | 5 | B | - | st-str | alf | i | es | o-a | 1.9 | me | ate |
| *Synedra famelica* Kützing | SynFam | 0 | 0 | 5 | P-B | - | str | alf | i | es | o | 1.3 | m | ats |

According to the CCA calculation results (Figure 9), it is possible to distinguish four groups of factors that determine the distribution of dominant species in the diatom communities across the Zarafshan River three upper sites. Figure 9a also included calculated indices SLA (Table 1) and WESI (Figure 8), whereas the triplot on Figure 9b was calculated for the abundant species only. The first group of factors included turbidity, salinity, oxygen demand, and pesticides, mostly affecting the community on the site Pakhtakor. Oil pollution associated with nitrates and copper and mainly expressed on the site Navoi. Interesting that only one species *Pleurosira laevis* was tolerant of detergent pollution, which comes from the middle part of the river. Pollution by phenols, zinc, and arsenic comes from the middle part of the river to the upper site Khatirchi and is opposite under the influence with the first group of factors that included turbidity and salinity. In the plot

of Figure 9a also can be seen that indices saprobity SLA and index WESI are not shown a specific reaction to any group of pollutants.

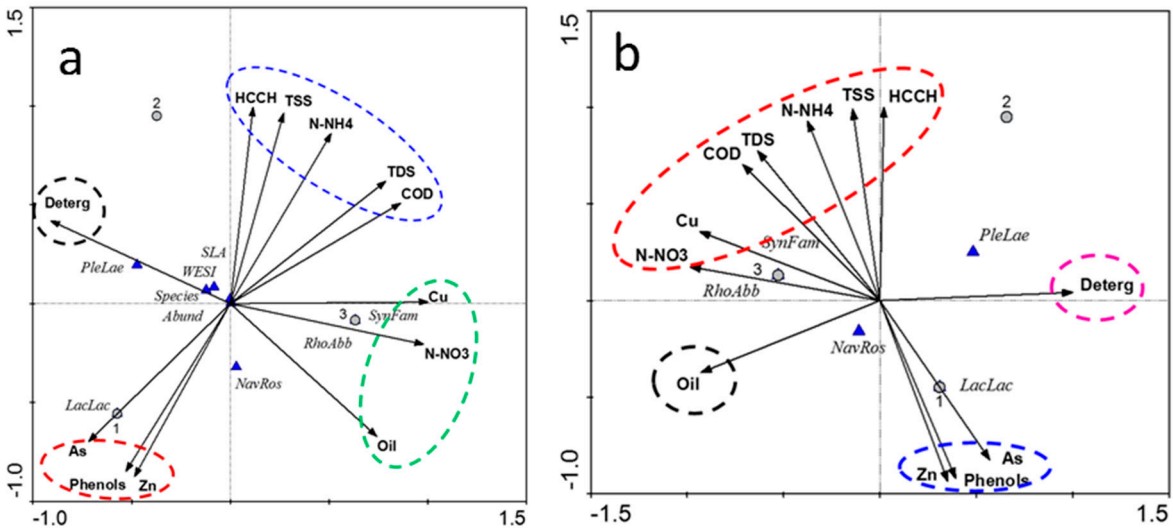

**Figure 9.** CCA plots of the three sites environmental variables and diatom species relationships in the Zarafshan River, 2009–2015; **a** — abundant species, total abundance scores, species richness, indices Sládeček, and WESI; and **b** — abundant species. Black arrows—environmental variables, blue triangles—diatom species, grey dots—sampling stations, dashed ovals—groups of environmental variables. 1, Khatirchi; 2, Pahtakor; 3, Navoi.

## 4. Discussion

### 4.1. Chemical Variables

According to the chemical data (Table 2), the content of heavy metals and heavy organic pollutants from the sources of industrial and agricultural pollution on the catchment basin were at a low level in the studied part of the lower Zarafshan River. We can see the pollutant distribution for the upper three sites only where coincide our collecting sites with the State monitoring points [12]. The average content of $N-NO_3$, $N-NO_2$, and $N-NH_4$ is in the maximum permissible concentrations of Class 6 established for the water quality classification system from an ecological point of view [1]. Therefore, nitrogen compounds, TSS, and BOD corresponded to level Class 5–6 of polluted water, increasing pollution from Khatirchi to Navoi. As a whole, the Zarafshan river water assessed from the upper part to the Navoi site was contaminated by nutrients due to agricultural pollution. Simultaneously, the main sources of the heavy metals (primarily arsenic and zinc) probably can be the ore-processed manufactures located in the territory of Tajikistan [18]. Similar pollution with a very high content of nutrients was found in the wastewater storage system in Kazakhstan [65], which can be properties of anthropogenically polluted water bodies in the semi-arid climatic zone of Central Asia [18,32].

### 4.2. Biological Variables

The study of the biological diversity of continental water bodies in the ecological aspect is an urgent modern scientific research trend. Studies of the species richness in Uzbekistan's rivers and reservoirs [23–28] revealed a high taxonomic diversity, amounting to about 2600 species and infraspecies algae, among which diatoms represented the richest group of 850 taxa. Since the general database has not yet been compiled for algae in Uzbekistan, which is to be done in the future, we cannot generally assess the participation of diatoms in algal flora. However, for individual water bodies, we can compare their share in the flora. Thus, in the Buzsuv Canal and the Chirchik River [66,67], 1562 species and infraspecies algae were identified, of which 659 were diatoms. As a result of our long-term studies of algae in the lower section of the Zarafshan River [68–77], its flora comprised 283 taxa, 198 of which were diatoms, that accounted for 70% of the list. It can be seen that

diatom species richness with 70% of algal flora in the studied lower part of the river was richest than in other river basins in Uzbekistan with 33% [23–28] and 40% [66,67]. Even in the upper part of the Zarafshan River [16], diatom algae represents 54% of the taxa list (223 from 415 taxa). Diatom percent increased up to 66% in the middle part of Zerafshan (218 from 331 taxa) [17]. Therefore, we can see increasing the role of diatom algae in the algal flora of the Zarafshan River from upper to lower parts about 1.3 times. The generic content also changed down the river from richest genera *Navicula* (30), *Nitzschia* (27), *Cymbella* (24) in the upper part to *Navicula* (36), *Nitzschia* (28), *Cymbella* (23) in the middle, and *Cymbella* (16), *Navicula* (15), *Nitzschia* (15) in the lower part communities. In comparison to the diatom species richness in the Pamir rivers in the upper part of the Amu Darya River can be seen that species-rich genera there are *Navicula*, *Nitzschia*, and *Pinnularia* (39 taxa in Pamir) [78], but *Pinnularia* species presented in the lower reaches of the Amu Darya communities by five species only.

Floristic comparison with statistical methods reveals three types of communities in the lower part of the Zarafshan River. Two upper sites can be marked by *Pleurosira laevis* as edificatory mesohalobic species described first for Uzbekistan and Central Asia [79]. The followed two sites communities were combined in the cluster with *Rhoicosphenia abbreviata* mesotrophic species domination. The third cluster represents communities with *Navicula rostellata* and *Pantocsekiella rossii*, planktonic oligotrophic species. That let us allow that water in the river channel changed with decreasing salinity and organic pollution down the river where plankton was most developed.

Our analysis shows that diatom species represent more than half of the riverine algae community, and their role increased from upper to lower reaches. Therefore, revealed diatoms can be used as indicators of the Zarafshan river environment because here they are rather represented and are the best aquatic inhabitants for bioindication [15,29–31].

### 4.3. Bioindication

The water's organic pollution in the Zarafshan River that was assessed with saprobity indicator species in the upper and middle parts [16,17] shows increasing organic pollution before our lower investigation of the river starts. Therefore, that demonstrated the problem, which a lower part of Zarafshan receives from the river and its catchment basin. The same dynamic of increasing organic pollution was demonstrated in Amu Darya's right tributary that slightly smaller than Zarafshan—the Qashkadarya [80,81]. The pollution increasing down the river was revealed in similar length rivers in Iran, Kazakhstan, and Tajikistan [82–85], close to Uzbekistan. Bioindication of other variables dynamic that we revealed with bioindication methods has never been studied in the Zarafshan river course. We revealed 91% of indicator taxa in the identified 198 species in the lower part of the river (Appendix A). Bioindication analysis of the nine indicator group distribution let us characterize the dynamic of diatom species' major ecological preferences in the lower part of Zarafshan. The results show the water oxygen saturation decreasing down the river with increasing salinity and turbidity. Acidification can be seen from site Navoi up to lower site Karakul. Organic pollution and the number of facultative heterotrophic and eutrophic species of Class 4 of water quality decreased down the river, but autotrophs were increased. As a whole, bioindication revealed pollution that receives the upper site Khatirchi by organic matter from the middle part of the river. The last part of the studied river course corresponds to Class 2 of water quality. The ecosystem index WESI revealed the negative influence of the river environment on the diatom communities' photosynthesis. Influence starts from the upper sites Khatirchi-Pahtakor-Navoi with the tendency of decreasing index that means increasing of impact. Simultaneously, the species richness and abundance increased down the river that, together with decreasing indices saprobity SLA, demonstrated the river ecosystem's self-purification. We constructed the ecological map that represents an important visualization instrument [33,83] recommended and applied in EU FWD [14,34] for the first time for Uzbekistan's rivers. The map in Figure 7 shows the fluctuation of water quality between Class 2 and 3 that correspond to the

groups of variables revealed by CCA, which influenced diatom communities in the different parts of the studied watercourse of Zarafshan. Ecological mapping can also reveal the seasonal fluctuation of indicator groups of water quality in the lower part of the Zarafshan River that can be analyzed concerning organic pollution [86].

## 5. Conclusions

We carried out a comprehensive environmental assessment of water quality and the aquatic ecosystem's state based on communities of diatoms for the first time for the lower reaches of the Zarafshan River. For the first time, we identified 198 species and subspecies of diatoms collected at six sites in 2009–2015. The indicator species (91% of the list) for the nine environmental variables reveal the dynamics of the river's water quality downstream, where organic pollution and overall toxic effects are not monitored yet. Analysis of the distribution of bioindicators, indices of organic pollution, and WESI and the statistics show an increase in salinity and turbidity and a decrease in organic pollution down the river. At the same time, the Navoi site is also a source of water acidification. The CCA shows that nutrients and heavy metals, and phenol pollution, enter the river from various sources. We found an increase in self-purification with an increase in the species richness and abundance of diatoms in the lower reaches of the Zarafshan. This indicates that the river's ecosystem successfully copes with the incoming pollution from the middle part of Zarafshan and its own catchment. For the first time in Uzbekistan, ecological mapping was applied to visualize problem areas of the river. This conclusion could not be made based on the chemical analysis of water only. This allowed us to recommend the inclusion of biological parameters and the expansion of state monitoring points up to the lower section of Karakul.

**Author Contributions:** Conceptualization, methodology, writing, editing, S.B.; investigation, K.M.; data curation, software, S.B.; formal analysis, K.M., S.B.; funding acquisition, S.B. All authors have read and agreed to the published version of the manuscript.

**Funding:** This research received no external funding.

**Institutional Review Board Statement:** Not relevant.

**Informed Consent Statement:** Not applicable for studies not involving humans or animals.

**Data Availability Statement:** Not applicable

**Acknowledgments:** The work was partly supported by the Israel Ministry of Aliyah and Integration.

**Conflicts of Interest:** The authors declare no conflict of interest.

## Appendix A

Taxonomical list of diatom algae with autecology and abundance scores of species revealed in the Zarafshan River, 2009–2015.

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
