# Peer review of "Diatom Algae-Indicators of Water Quality in the Lower Zarafshan River, Uzbekistan"

_water, doi:10.3390/w13030358_

Round 1

Reviewer 1 Report

The manuscript presented for review concerns Diversity of diatom algae as bioindicators of water quality in the Zarafshan River, Uzbekistan. This is a topic related to water quality. It is a current topic, undertaken by many researchers, and therefore not completely new. However, this does not change the fact that this topic is important and requires in-depth research and observation. It also fits in with the recommendations for monitoring the aquatic environment. It is clear that the authors approached the research fairly and with appropriate scientific knowledge. The manuscript provides a lot of information.

However, the manuscript contains errors that must be corrected in order for it to be processed further.

The manuscript needs thorough linguistic improvement. It contains numerous linguistic and stylistic errors. In some places it is difficult to understand what the Authors mean. I recommend checking the linguistic correctness by a good translator and / or native speaker.

Introduction - It is somewhat confusing to start the introduction with an extensive description of the Amu Darya River, which is not directly studied. The information provided is too detailed and unnecessary. In addition, they give the impression that the authors have given the manuscript an inappropriate title.

Fig.1 - is of poor quality, which makes it difficult to read.

Fig. 4b - what specific temperature values correspond to the terms used by the authors? What is e.g. "warm water"? After all, this term may have different values in different parts of the world.

Therefore, I recommend that the manuscript be seriously improved.

Author Response

Responses to Reviewer 1

Dear Reviewer,

Thank you for comments and notes. I corrected ms follow by your comments. Below you can find my responses to each comment.

The manuscript needs thorough linguistic improvement. It contains numerous linguistic and stylistic errors. In some places it is difficult to understand what the Authors mean. I recommend checking the linguistic correctness by a good translator and / or native speaker. Response: corrected

Introduction - It is somewhat confusing to start the introduction with an extensive description of the Amu Darya River, which is not directly studied. The information provided is too detailed and unnecessary. In addition, they give the impression that the authors have given the manuscript an inappropriate title. Response: corrected

Fig.1 - is of poor quality, which makes it difficult to read. Response: improved, replaced

Fig. 4b - what specific temperature values correspond to the terms used by the authors? What is e.g. "warm water"? After all, this term may have different values in different parts of the world. Response: added to the MM section

With best regards,

Prof Sophia Barinova,

corresponding author

Reviewer 2 Report

A comprehensive assessment of the ecological state of the lower reaches of the Zarafshan River using bioindication of water quality by diatoms based on species' ecological preferences, pollution indices, statistics, and ecological mapping was conducted in 2009-2015. The total of 198 diatom species and infraspecies from 195 samples were collected and Navicula, Nitzschia, and Cymbella were identified as the richest genera. Salinity and turbidity were proved as the important parameters. An increase in self-purification with increasing species richness and abundance of diatom algae in the lower part of Zarafshan was verified. An increase in self-purification processes with the adequate water use was also demonstrated. This work is still crude and need to be polished.

  1. The introduction is not charming and just like a case, not a scientific introduction.
  2. There's some confusion about logic. no diversity could be found in this manuscript, no climate change data could be found in this manuscript. But the title and the conclusion meant the diversity and climate change, respectively.

  3. statistical methods should be mentioned in the section of materials and methods.
  4. some figures is crude and need to be polished.
  5. Conclusions is too long and reference can't appear in this section.

Author Response

Responses to Reviewer 2

Dear Reviewer,

Thank you for comments and notes. I corrected ms follow by your comments. Below you can find my responses to each comment.

  1. The introduction is not charming and just like a case, not a scientific introduction. Response: rephrased, shortened and the relevant references added
  2. There's some confusion about logic. no diversity could be found in this manuscript, no climate change data could be found in this manuscript. But the title and the conclusion meant the diversity and climate change, respectively. Response: added in MM section data about climatic condition and its change down the river and in time.
  3. statistical methods should be mentioned in the section of materials and methods. Response: added
  4. some figures is crude and need to be polished. Response: replaced
  5. Conclusions is too long and reference can't appear in this section. Response: shortened

With best regards,

Prof Sophia Barinova,

corresponding author

Reviewer 3 Report

This manuscript assessed the water quality of the lower reaches of the Zarafshan River by diatoms species biodiversity, pollution indices and ecological mapping. This work presented that diatoms can be good indicators of water quality in this river. It should be a valuable contribution to the literature. I could recommend this manuscript. However, the manuscript should be revised a lot before publication:

  1. There is no description of diatom in the Introduction, for example, what are the advantages of diatoms as environmental indicators, how to apply diatom in river water quality assessment...
  2. Line 133-134: ” ...Index values SLA ranges from 1 to 4 for the aquatic environment...”. I guess 1=no pollution and 4=very pollution? This should be explicitly stated.
  3. Line 140: “The River Pollution Index (RPI), according to Sumita [21], ” what is the range of RPI Index values, and how does the RPI Index value indicate the state of water pollution? this should be explicitly stated.
  4. Line 150: ”...We have chemical information from three upper monitoring points only...” Will the lack of data from the last three points affect the result analysis?
  5. Table 1:“No of Species”should be changed to“Number of species”.
  6. Line 165:” “3.2 Biological characteristics of the lower part of the Zarafshan River.” The relative richness of dominant species should be presented, and these data should be supported by statistical analysis.
  7. Line 166: “Altogether 198 taxa of diatoms were ...”, How many genera do these 198 taxa belong to?
  8. Line 198: “3.3 Bioindicators in the lower part of the Zarafshan River.” The authors should not only describe the figures simply, but provide the detail percentage of each group which is hard to get from the figure.
  9. Line 205: Please explain in detail the results illustrated in Figure 4d.
  10. Line 242-243: “Figure 6 shows that the identified species indicators of organic pollution correspond to four Classes of water quality.”, What are the main diatom species of the identified species indicators of organic pollution?
  11. Line 260-261: How to demonstrate self-purification of the river water?
  12. Figure 8: Explain what the red (NORM) line means?
  13. Line 367: “We revealed 91% of indicators from the identified 198 species in the lower part of the river.” The author should provide the detail information of these indicators, and which kind of environment they could indicate.
  14. Line 454: ”327” should be changed to“327 p.”
  15. Line 458: ”555585” should be changed to“pp. 070-072”
  16. Line 501: ”Encyonema part., Encyonopsis und Cymbellopsis.” should be changed to“Encyonema part., Encyonopsis und Cymbellopsis.”
  17. I recommend a careful cross-check of the references before to resubmit the manuscript.

Author Response

Responses to Reviewer 3

Dear Reviewer,

Thank you for comments and notes. I corrected ms follow by your comments. Below you can find my responses to each comment.

  1. There is no description of diatom in the Introduction, for example, what are the advantages of diatoms as environmental indicators, how to apply diatom in river water quality assessment... Response: added reference
  2. Line 133-134: ” ...Index values SLA ranges from 1 to 4 for the aquatic environment...”. I guess 1=no pollution and 4=very pollution? This should be explicitly stated. Response: done
  3. Line 140: “The River Pollution Index (RPI), according to Sumita [21], ” what is the range of RPI Index values, and how does the RPI Index value indicate the state of water pollution? this should be explicitly stated. Response: done
  4. Line 150: ”...We have chemical information from three upper monitoring points only...” Will the lack of data from the last three points affect the result analysis? Response: We describe the influence of environmental conditions for the upper 3 sites only and recommend to expand state monitoring points to the lower part of the river up to the Karakul region.
  5. Table 1:“No of Species”should be changed to “Number of species”. Response:  done
  6. Line 165:” “3.2 Biological characteristics of the lower part of the Zarafshan River.” The relative richness of dominant species should be presented, and these data should be supported by statistical analysis. Response:  Added to section CCA
  7. Line 166: “Altogether 198 taxa of diatoms were ...”, How many genera do these 198 taxa belong to? Response: done
  8. Line 198: “3.3 Bioindicators in the lower part of the Zarafshan River.” The authors should not only describe the figures simply, but provide the detail percentage of each group which is hard to get from the figure. Response: done
  9. Line 205: Please explain in detail the results illustrated in Figure 4d. Response: added to MM part
  10. Line 242-243: “Figure 6 shows that the identified species indicators of organic pollution correspond to four Classes of water quality.”, What are the main diatom species of the identified species indicators of organic pollution? Response: added
  11. Line 260-261: How to demonstrate self-purification of the river water? Response: by decreasing of calculated index saprobity SLA.
  12. Figure 8: Explain what the red (NORM) line means? Response: done
  13. Line 367: “We revealed 91% of indicators from the identified 198 species in the lower part of the river.” The author should provide the detail information of these indicators, and which kind of environment they could indicate. Response: represent in Appendix 1 and MM part, references for each ecological group environmental ranges added
  14. Line 454: ”327” should be changed to“327 p.” Response: done
  15. Line 458: ”555585” should be changed to“pp. 070-072” Response: done
  16. Line 501: ”Encyonema part., Encyonopsis und Cymbellopsis.” should be changed to“Encyonema part., Encyonopsis und Cymbellopsis.” Response: done
  17. I recommend a careful cross-check of the references before to resubmit the manuscript. Response: done

With best regards,

Prof Sophia Barinova,

corresponding author

Round 2

Reviewer 1 Report

The reviewers' comments were largely applied and the manuscript revised. However, the introduction still needs improvement - the authors initially describe Amu Darya in detail, which is not a direct object of research. This misleads the reader and renders the manuscript unfinished.

Author Response

Response to the Reviewer 1

Dear Editor and the Reviewer,

Thank you for comments.

The reviewers' comments were largely applied and the manuscript revised. However, the introduction still needs improvement - the authors initially describe Amu Darya in detail, which is not a direct object of research. This misleads the reader and renders the manuscript unfinished.

Response: Introduction reweighted

Reviewer 2 Report

The introduction is still need to be polished and the research hypothesis should be mentioned.

No climate data in this manuscript could be found.

The conclusion is still very long and need to be shortened.

Author Response

Response to the Reviewer 2

Dear Editor and the Reviewer,

Thank you for your comments.

The introduction is still need to be polished and the research hypothesis should be mentioned. Response: done

No climate data in this manuscript could be found. Response: Description of climate data can be found in the part 2.1. Description of study site

The conclusion is still very long and need to be shortened. Response: done

This manuscript is a resubmission of an earlier submission. The following is a list of the peer review reports and author responses from that submission.